# The Influence of Micro-Hexapod Walking-Induced Pose Changes on LiDAR-SLAM Mapping Performance

**DOI:** 10.3390/s24020639

**Published:** 2024-01-19

**Authors:** Hiroshi Seki, Yuhi Yamamoto, Sumito Nagasawa

**Affiliations:** 1Mechanical Engineering, Graduate School of Engineering and Science, Shibaura Institute of Technology, 3-7-5 Toyosu, Koto-ku, Tokyo 135-8548, Japan; md22049@shibaura-it.ac.jp (H.S.); md22112@shibaura-it.ac.jp (Y.Y.); 2Department of Engineering Science and Mechanics, College of Engineering, Shibaura Institute of Technology, 3-7-5 Toyosu, Koto-ku, Tokyo 135-8548, Japan

**Keywords:** SLAM, LiDAR, hexapod robot, crawler robot, micro-hexapod

## Abstract

Micro-hexapods, well-suited for navigating tight or uneven spaces and suitable for mass production, hold promise for exploration by robot groups, particularly in disaster scenarios. However, research on simultaneous localization and mapping (SLAM) for micro-hexapods has been lacking. Previous studies have not adequately addressed the development of SLAM systems considering changes in the body axis, and there is a lack of comparative evaluation with other movement mechanisms. This study aims to assess the influence of walking on SLAM capabilities in hexapod robots. Experiments were conducted using the same SLAM system and LiDAR on both a hexapod robot and crawler robot. The study compares map accuracy and LiDAR point cloud data through pattern matching. The experimental results reveal significant fluctuations in LiDAR point cloud data in hexapod robots due to changes in the body axis, leading to a decrease in map accuracy. In the future, the development of SLAM systems considering body axis changes is expected to be crucial for multi-legged robots like micro-hexapods. Therefore, we propose the implementation of a system that incorporates body axis changes during locomotion using inertial measurement units and similar sensors.

## 1. Introduction

In recent years, significant strides have been made in the research and development of LiDAR-based simultaneous localization and mapping (LiDAR-SLAM), particularly in its application to disaster exploration robots. However, to our knowledge, no prior research has explored the implementation of LiDAR-SLAM on micro-hexapods. This study conducts a comparative experiment involving LiDAR-SLAM mapping using a hexapod robot and small-scale crawler robot to obtain maps. We quantitatively assess the impact of micro-hexapod locomotion on LiDAR-SLAM by processing map data through image processing techniques.

Several robots designed for hazardous environment exploration have been developed in recent years [1]. These robots offer a safe and efficient alternative to human exploration. The advent of SLAM technology has played a pivotal role in creating highly accurate maps of the surrounding environment and providing precise self-position estimation, leading to increased utilization in disaster exploration robots [2]. SLAM can be categorized into various types, such as V-SLAM, LiDAR-SLAM, and Depth SLAM, depending on the sensors used. Among these, LiDAR-SLAM stands out for its high accuracy over a wide area [3]. Despite its capability to generate highly accurate map data based on high-density point-cloud data, LiDAR-SLAM is complex and computationally intensive, leading to accumulated position measurement errors and deviations from the true position. Efforts have been made to reduce the computational load and enhance the accuracy of LiDAR-SLAM [4,5].

Moreover, various types of robots designed for exploration purposes have been developed [1]. Among these, multi-legged robots, known for their superior mobility [6], have been proposed for exploration due to their ability to navigate rough terrains effectively. The micro-hexapod [7,8,9,10,11,12,13,14] is a robot that combines elements of micro-robots [15,16,17,18,19] and hexapod robots [20,21,22,23], the latter being robots with six legs. Micro-robots are particularly appealing for their suitability in confined spaces, cost-effectiveness, and potential for mass production. Leg-type robots, especially hexapod robots, can walk stably without constant posture control [23,24,25,26].

The introduction of LiDAR-SLAM to micro-hexapods is expected to enhance their capabilities as disaster exploration robots. However, there has been a lack of quantitative research evaluating the map information and self-positioning accuracy achieved by implementing LiDAR-SLAM on micro-hexapods. Previous studies explored SLAM implementation in hexapod robots [27,28,29,30]. However, these studies did not evaluate the differences between various locomotion mechanisms, such as crawler robots and hexapod robots. These studies operated under the assumption that SLAM could perform with the same level of accuracy on a hexapod robot as on robots with different locomotion methods. In this context, our study aims to evaluate the accuracy of mapping estimation by several types of SLAM on a hexapod robot. In practice, leg movements differ from the continuous ground contact typical of other robots, introducing factors such as tilt, sway, and vibration due to leg movements. Therefore, it is essential to quantitatively evaluate the impact of leg movement on SLAM in the context of hexapod robots.

We experimentally investigated the effect of leg movement on LiDAR-SLAM in a small hexapod robot. Experiments involved both a small hexapod robot and crawler-type robot, generating maps in each case. The experiment was conducted in a simple environment to disregard the impact of environmental complexity on the map and measure only the impact of the robot’s body axis changes on the map. We compared and evaluated the similarity between the ideal map data and map data obtained in our experiments. This study aims to investigate the influence of hexapod robot leg mobility mechanisms on SLAM through controlled experiments involving two robots with distinct mobility mechanisms. We employed Hu moment invariants for pattern matching to compare and evaluate maps. The results of this comparison will allow us to quantitatively measure the impact of leg movement on LiDAR-SLAM in hexapod robots. Based on our findings, we intend to propose a system for future LiDAR-SLAM on micro-hexapods.

This paper consists of four sections. In Section 1, we describe the research background regarding SLAM installed in micro hexapods and the purpose of this paper. Section 2 details the hardware and software configurations of the hexapod and crawler robot used in the experiment and describes how to evaluate the data obtained from the SLAM experiment. Section 3 presents the results of the map data and LiDAR point cloud data obtained in the SLAM experiment and presents the results of pose fluctuations during the walking of the hexapod and crawler robot. Section 4 presents the conclusion of this paper.

## 2. Materials and Methods

### 2.1. Methodology Overview

In this study, we conducted a controlled experiment in which we implemented LiDAR-SLAM using two robots that differ solely in their locomotion mechanisms. The primary goal is to investigate the impact of different movement mechanisms on SLAM while maintaining consistency across all other components. To facilitate this controlled experiment, it was imperative to utilize identical components (computers, LiDAR, etc.) across both robots, with the exception being the locomotion mechanism. Additionally, we considered potential scale effects when reducing the size. Therefore, both robots were scaled down to match the dimensions of a micro-hexapod. However, it is worth noting that, at the current state of technology, micro-hexapods smaller than 10 cm in size are still in the research and development stage, making it challenging to introduce LiDAR-SLAM to them. Consequently, we designed and constructed a small hexapod robot and a compact crawler robot specifically for use in our experiments.

In this experiment, we employed Hector-SLAM, a two-dimensional (2D) LiDAR-SLAM approach utilizing scan matching with LiDAR point cloud data. Our objective was to create a 2D environmental map exclusively using the point cloud data acquired by the 2D LiDAR. Subsequently, we evaluated the resulting 2D environmental map through pattern matching—a method used to measure the similarity between images and identify objects. To assess the comparison between ground truth and measurement data, we utilized the Hu moment invariants method for pattern matching.

### 2.2. Robot Specifications

In this study, we created a hexapod robot and a crawler robot, illustrated in Figure 1a,b, with their specifications summarized in Table 1. Key shared components include the computer board and LiDAR system. The computer board, a Raspberry Pi 4 (8 GB), is a compact device measuring 86 × 55 mm, serving to control the robot’s movements during LiDAR-SLAM operations. For LiDAR, we employed RPLIDAR’s A1M8 two-dimensional LiDAR—a mechanical device capable of 360° rotation and producing 2D maps.

#### 2.2.1. Hexapod Robot

The compact hexapod robot, as depicted in Figure 1a, features Tower Pro’s SG92R micro servo motor as the leg actuator, as shown in Figure 2a. These micro servo motors, with dimensions of 23 × 12.2 × 27 mm and a weight of 9 g, provide a maximum torque of 2.5 kgf-cm. The completed hexapod leg is presented in Figure 2b.

To minimize the leg size for efficient walking, we designed the hexapod with six legs, each having two degrees of freedom. A parallel link mechanism controlled the vertical movement of the legs, distributing force from one axis of the servomotor to the other, with a link length of 35 mm. Given that the robot’s weight, excluding the legs, is approximately 1 kg, each leg needed to support at least 0.33 kgf to bear the weight of the robot with three legs. The maximum vertical output of this leg was 0.71 kgf, providing an additional force approximately twice as much as needed.

The hardware configuration of the hexapod robot is shown in Figure 3. The Raspberry Pi manages the robot movement and transmits LiDAR data to ROS topics. The data are then sent to the host computer via Wi-Fi communication for LiDAR-SLAM processing. The Raspberry Pi also governs the servo motors through I2C communication with the servo motor driver.

The node configuration of the hexapod robot is depicted in Figure 4. On the host PC, we execute roscore, hector_slam_launch, range_data_csv, and hexapod_controller. Subsequently, on the Raspberry Pi mounted on the robot, we run rplidar_ros and hexapod_ros. LiDAR data are collected by rlidar_ros and transmitted to hector_slam_launch and range_deta_csv via topic communication. Action commands to the hexapods are sent from hexapod_controller to hexapod_ros through Topic communication to control the movement of the legs.

#### 2.2.2. Crawler Robot

We constructed a compact crawler-type robot, as depicted in Figure 1b, similar in size to the hexapod robot. This small crawler robot was designed with an Arduino as its primary controller, measuring 10 × 10 cm and weighing approximately 280 g, inclusive of batteries.

For the crawler part of our crawler robot, we used the ZUMO shield shown in Figure 5. The ZUMO shield is a product designed to control ZUMO by attaching an Arduino UNO microcontroller board. For this application, the GPIO pins of the Raspberry Pi are directly interfaced with the Arduino, allowing the Raspberry Pi to govern the device without relying on the Arduino.

The hardware configuration of the crawler robot is illustrated in Figure 6. Similar to the hexapod robots, the Raspberry Pi manages the robot motion and transmits LiDAR data within ROS topics. Data transmission from the Raspberry Pi to the host computer occurs through Wi-Fi communication, and LiDAR-SLAM processing is carried out on the host PC. Motor control is established by connecting the GPIO pins of the Raspberry Pi to the pins on the ZUMO shield, transmitting PWM signals to the motor driver within the ZUMO.

The node configuration of the crawler robot is presented in Figure 7. As with the hexapod robots, we execute roscore, hector_slam_launch, range_data_csv, and teleop_node on the host PC. On the Raspberry Pi mounted on the robot, we run rplidar_ros and zumo. Like the hexapod robots, LiDAR data are collected by rplidar_ros and sent to hector_slam_launch and range_deta_csv via topic communication. Action commands to the crawler robot are transmitted from teleop_node to zumo through Topic communication to control the robot’s movements.

### 2.3. SLAM

In this study, we will employ Heterogeneous Cooperating Terrain-based Outdoor Robot SLAM (Hector-SLAM) as our 2D-SLAM system. Hector-SLAM, developed by the “Hector” robotics research team at the Technical University of Darmstadt, Germany, is specifically designed for outdoor environments. The system operates based on a grid map and relies on scan matching as its fundamental principle.

Hector-SLAM facilitates mapping and self-position estimation exclusively using LiDAR scan data. While it can achieve enhanced SLAM precision when combined with additional inputs such as odometry and inertial measurement unit (IMU) data, we exclusively utilize pure LiDAR scan data for mapping and assess the impact. Therefore, we have chosen Hector-SLAM, an open-source SLAM system capable of self-position estimation solely from LiDAR data, to perform SLAM using LiDAR exclusively.

### 2.4. Evaluation Method

In this experiment, we utilize Hu moment invariants [31] to evaluate the similarity between the generated map and the ground truth, as well as the resemblance between temporally adjacent LiDAR point cloud data images. Hu moment invariants are selected due to their inherent properties of invariance with respect to translation, size, and rotation, making them an appropriate choice for evaluating shape similarity.

As per the definition provided in [32], the 2D moment is expressed as follows:(1)$$m_{pq}=\int_{-\infty }^{\infty }\int_{-\infty }^{\infty }x^{p}y^{q}f\left(x,y\right)dxdy$$
p,q=0, 1, 2,…
where f(x,y) is the density distribution function. Applying this to digital images, we obtain the following:(2)$$m_{pq}=\int_{-\infty }^{\infty }\int_{-\infty }^{\infty }x^{p}y^{q}I(x,y)$$

The central moment $\mu_{pq}$ is defined as follows:(3)$$\mu_{pq}=\sum_{x}\sum_{y}{\left(x-\bar{x}\right)}^{p}({y-\bar{y})}^{q}I(x,y)$$
where
(4)$$\bar{x}=\frac{m_{10}}{m_{00}},\;\bar{y}=\frac{m_{01}}{m_{00}}\;$$

The central moment is invariant to the translation of the image. To achieve scale invariance, we employ normalization, and the normalized central moment $\mu_{pq}$ is defined as follows:(5)$$\mu_{pq}=\frac{\mu_{pq}}{\mu_{00}^{\gamma }}$$

Normalized central moments yield the following seven distinct moments. In the context of a mirror image, the seventh moment undergoes a change in sign.
(6)$$\;H_{1}=\mu_{20}+\mu_{02}$$
(7)$$\;H_{2}={\left(\mu_{20}+\mu_{02}\right)}^{2}+4\mu_{11}^{2}$$
(8)$$\;H_{3}={\left(\mu_{30}-3\mu_{12}\right)}^{2}+{\left(3\mu_{21}-\mu_{03}\right)}^{2}$$
(9)$$\;H_{4}={\left(\mu_{30}+\mu_{12}\right)}^{2}+{\left(\mu_{21}+\mu_{03}\right)}^{2}$$
(10)$$H_{5}=\left(\mu_{30}-3\mu_{12}\right)\left(\mu_{30}+\mu_{12}\right)\left[{\left(\mu_{30}+\mu_{12}\right)}^{2}-3{\left(\mu_{21}+\mu_{03}\right)}^{2}\right]+\left(3\mu_{21}-\mu_{03}\right)\left(\mu_{21}+\mu_{03}\right)[3{\left(\mu_{30}+\mu_{12}\right)}^{2}-{\left(\mu_{21}+\mu_{03}\right)}^{2}]$$
(11)$$\;H_{6}=\left(\mu_{20}-\mu_{02}\right)\left[{\left(\mu_{30}+\mu_{12}\right)}^{2}-{\left(\mu_{21}+\mu_{03}\right)}^{2}\right]+4\mu_{11}\left(\mu_{30}+\mu_{12}\right)\left(\mu_{21}+\mu_{03}\right)$$
(12)$$H_{7}=\left(3\mu_{21}-\mu_{03}\right)\left(\mu_{30}+\mu_{12}\right)\left[{\left(\mu_{30}+\mu_{12}\right)}^{2}-3{\left(\mu_{21}+\mu_{03}\right)}^{2}\right]-\left(\mu_{30}-3\mu_{12}\right)\left(\mu_{21}+\mu_{03}\right)[(3{\left(\mu_{30}+\mu_{12}\right)}^{2}-{\left(\mu_{21}+\mu_{03}\right)}^{2}]$$

Let the two contours to be compared be contours A and B, and we define Hu moments as: $H_{i}^{A}$, $H_{i}^{B}(i=1,2,\ldots 7)$.

The norm D(A, B) between contours A and B is expressed as follows.
(13)$$D\left(A,\;B\right)=\sum_{i=1}^{7}\left|\frac{1}{m_{i}^{A}}-\frac{1}{m_{i}^{B}}\right|$$
where
(14)$$m_{i}^{A}=sign\left(H_{i}^{A}\right)log(H_{i}^{A})$$

Define the norm D(A, B) as the similarity. The smaller the similarity value, the greater the similarity between the shapes of the two images. In this paper, the above evaluation is performed using OpenCV’s matchShapes() function.

## 3. Results and Discussion

In this research, two experiments were conducted.

In Section 3.1, an experiment was conducted to evaluate the extent to which the magnitude of the body axis change during movement varies depending on the difference in the robot’s movement mechanism. The small hexapod robot and crawler robot were equipped with the same IMU to estimate the robot’s posture during movement.

In Section 3.2, based on the fact that the magnitude of the body axis changes depending on the difference in the robot’s moving mechanism, we performed SLAM on the small hexapod robot and the crawler robot and evaluated the generated maps and LiDAR point cloud data during the SLAM execution.

### 3.1. Robot Body Axis Changes

The experiment was conducted to evaluate the extent to which body axis changes during locomotion are affected by differences in the robot’s locomotion mechanism. The small hexapod robot and crawler robot were placed in a straight line on the flat table shown in Figure 8. The robot was placed in a straight line on the flat table. The running time of the hexapod robot was long enough to measure the periodic motion of the crawler robot, and the same running time was used for the crawler robot.

The angular velocity of each robot was measured using the same IMU during running, and the roll and pitch angles were estimated by integrating the angular velocity. Figure 9a shows the posture variation of the hexapod robot while running, and Figure 9b shows the posture variation of the crawler robot.

Figure 9a,b show that the hexapod robot undergoes periodic posture fluctuations during running, accompanied by periodic changes in the roll and pitch angles of its body axes.

### 3.2. SLAM Experimental Results

The experiment was conducted in an indoor environment, as depicted in Figure 10a. Mapping using Hector-SLAM was performed five times for both the hexapod and crawler robots. To serve as a reference for assessing the similarity with the generated maps, we established the ground truth map data for the physical environment, depicted in Figure 10b. In Figure 10b, white areas represent passable spaces, black areas are impassable objects, and gray areas are unknown areas.

#### 3.2.1. Traveled Path

Figure 11a shows the traveled path of the hexapod robot and Figure 11b shows the path of the crawler robot. Figure 11c,d are enlarged views of the traveled paths of the first trial for the hexapod and crawler robot.

From Figure 11c, the traveled path of the hexapod robot has a blur width. Note that in Figure 11c,d, the number of self-position estimation plots for the hexapod robot is large because the speed of the hexapod robot is smaller than that of the crawler robot, and the period of self-position estimation is constant.

#### 3.2.2. Map Generated by SLAM

Table 2 presents images of maps generated by both the hexapod and crawler robots, along with their corresponding similarity when compared to the ground truth.

Observations from the first and third trials of the hexapod robot reveal that maps generated by this robot sometimes extend beyond the boundaries of the ground truth. This phenomenon can be attributed to the tilt of the LiDAR sensor, caused by shifts in the hexapod robot’s body axis, resulting in the detection of voids beneath the wall shown in Figure 10a. This, in turn, leads to substantial fluctuations in the LiDAR point cloud data. It is important to note that the protruding regions in the first and third trials of the hexapod robot are considered as contours independent of the map and have been excluded from the similarity evaluation.

As indicated in Table 2, the average similarity of the maps generated by the hexapod robot is 0.0900, whereas for the crawler robot, it is 0.0677. The value for the hexapod is 1.33 times higher than that for the crawler robot. This demonstrates that employing a hexapod robot for map generation results in maps with shapes differing from the physical environment in comparison to the use of a crawler robot.

#### 3.2.3. Point Cloud Data Acquired by LiDAR

At a specific point in the experiment, we generated LiDAR point cloud data images, as displayed in Figure 12, by connecting adjacent points from the LiDAR point cloud data with straight lines. Subsequently, we replicated this image generation process each time the scan topic, representing LiDAR point cloud data, was published within the ROS system. This allows us to continuously assess the similarity between temporally adjacent LiDAR point cloud data images, as illustrated in Figure 13. The assessment of the similarity between these images was performed continuously from the commencement of the robot movement to its conclusion. This process was repeated for each trial of the experiment, which included 5 trials for the hexapod robot and 5 trials for the crawler robot, for a total of 10 trials. This results in the creation of box plots and their magnified versions, as illustrated in Figure 14a,b, respectively. The cross marks in the boxplot mean the average value, and the plotted dots mean that they were calculated as outliers. Furthermore, we utilized the Steel–Dwass method to determine whether there were significant differences among the 10 groups. The Steel–Dwass method was performed with JMP 16.0 (SAS Institute, Cary, NC, USA).

From Figure 14a,b, it is evident that the hexapod robot exhibits higher average and median values for the similarity between temporally adjacent LiDAR point cloud data images when compared to the crawler robot. The mean value for the crawler robot is approximately 0.01, whereas the mean value for the hexapod ranges from 0.04 to 0.13. The median value with the hexapod is approximately two times higher than that with the crawler robot.

This observation suggests that the hexapod robot introduces significant fluctuations in point cloud data each time it acquires LiDAR point cloud data. Moreover, Figure 14a highlights a greater presence of outliers in the case of the hexapod robot in comparison to the crawler robot, indicating more frequent extreme fluctuations in point cloud data. This is attributed to the hexapod robot’s LiDAR tilting due to changes in its body axis, leading to the detection of voids beneath the wall in Figure 10a and significant variations in the LiDAR point cloud data.

Furthermore, Figure 14b illustrates that as the experiments progressed, the hexapod robot group exhibited stair-step increasing values for the third quartile and mean in the box plot. This can be attributed to the diminishing battery power over the course of the experiments, resulting in reduced leg-holding torque and greater body axis variations during walking.

The *p*-values obtained for each combination using the Steel–Dwass method are presented in Figure 15. We initially expected that comparisons between robots of the same type would not yield significant differences, and cases contrary to our expectations are highlighted in bold.

Examining Figure 15, it is evident that the distribution of similarity groups between the hexapod and the crawler robots results in *p*-values below 0.05 for all combinations, signifying statistical significance. Consequently, it is established that the hexapod and crawler robots exhibit varying degrees of variability in LiDAR point cloud data.

Furthermore, in regard to the distribution of similarity groups among the crawler robots, there is no statistically significant difference in the distribution of similarity for LiDAR point cloud data images, except for the first trial. In contrast, the hexapod robot exhibits statistical significance in all combinations, with the exception of the second and third trials. Thus, this indicates that the hexapod robot’s similarity distribution varies from trial to trial. This is presumed to be a result of the depletion of battery power during the experiments, leading to diminished leg-holding torque and increased body axis variations during walking.

This experiment highlights that the hexapod robot experiences reduced accuracy in the generated maps through SLAM and decreased precision in the acquired LiDAR point cloud data when compared to the crawler robot. These effects are attributed to the tilting of the LiDAR sensor induced by changes in the body axis. This tilting causes the LiDAR to deviate from the horizontal plane, resulting in measured distances exceeding their true values. It is suggested that this issue could potentially be addressed by utilizing an IMU to detect variations in the body axis of the hexapod robot and the LiDAR sensor, enabling the correction of LiDAR point cloud data values based on the robot’s pitch and yaw angles.

A limitation of this study is that it does not explore the impact of walking types other than hexapod walking on SLAM. Bipedal and quadrupedal walking might present challenges due to the need for dynamic balance. Multipedal walking with eight or more legs, although maintaining a stable body axis, could be heavily influenced by the terrain when the robot drags its body.

## 4. Conclusions

This study focuses on evaluating the influence on maps obtained by implementing a self-location estimation system, 2D LiDAR-SLAM, on a micro-hexapod. Maps were generated using 2D LiDAR-SLAM, with a small hexapod robot mimicking a micro-hexapod and small crawler-type robot. The evaluation of the similarity between ideal and experimental map data was conducted using pattern matching with Hu moment invariants.

The experimental data indicate that the hexapod robot is more sensitive to small noises and sudden outliers in the map compared to a typical crawler robot, due to body axis changes resulting from the walking motion shown in Section 3.1. The average similarity of maps generated by the crawler robot is 0.0677, whereas for the hexapod robot, it is 0.0900, yielding an error that is 1.33 times larger. The magnitude of change between temporally adjacent LiDAR point cloud data was evaluated; the hexapod robot is found to exhibit larger changes, and significant differences are observed between the hexapod and crawler robots, as assessed by the Steel–Dwass method.

This study highlights that the dynamic characteristics of the micro-hexapod impact the accuracy of LiDAR-SLAM maps. As a future scope of this study, the development of a LiDAR-SLAM system suitable for the micro-hexapod is expected. The use of an IMU to detect changes in the body axis of the hexapod robot and LiDAR sensor, along with the correction of LiDAR point cloud data values, is planned. Additionally, it is imperative to develop a lightweight SLAM system capable of expanding the field of view and enabling 3D mapping by utilizing the swaying of the robot body axis during walking.

## Figures and Tables

**Figure 1 sensors-24-00639-f001:**
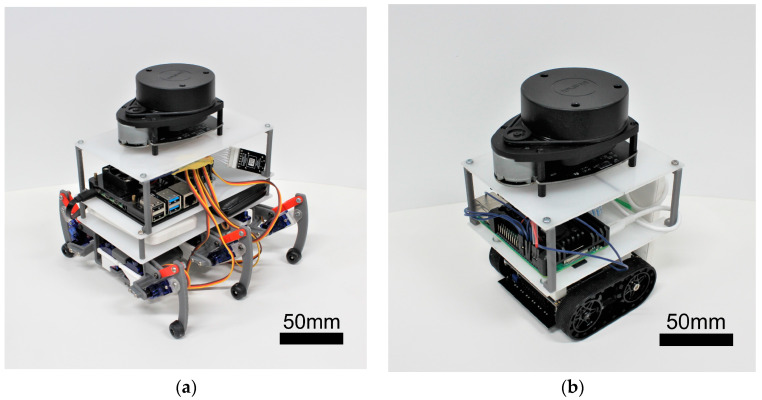
Images of robots: (**a**) hexapod robot; (**b**) crawler robot.

**Figure 2 sensors-24-00639-f002:**
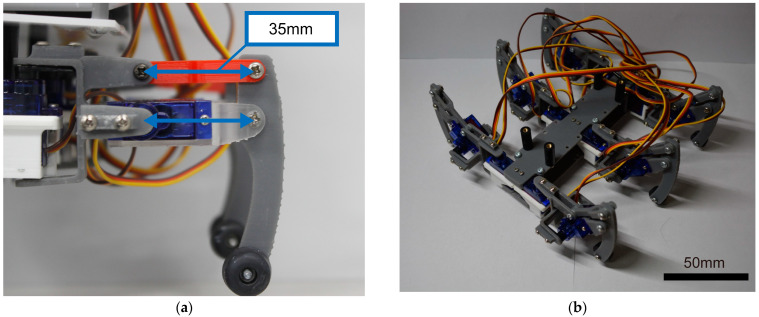
Image of the robot: (**a**) a leg of one of the hexapods; (**b**) legs of the hexapods.

**Figure 3 sensors-24-00639-f003:**
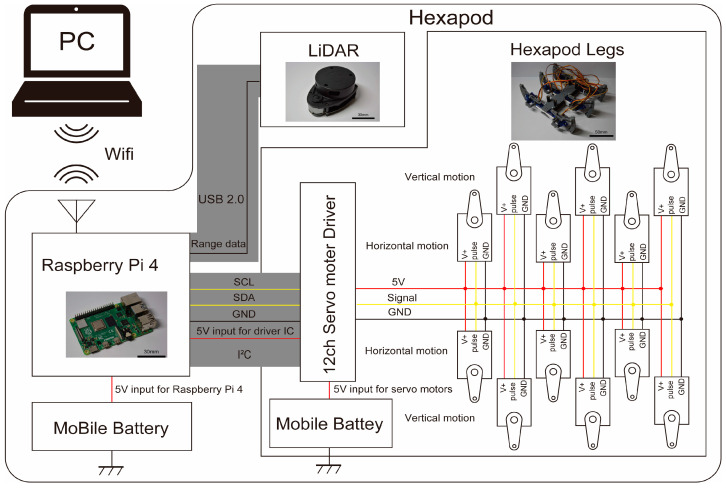
Architecture of the hexapod robot.

**Figure 4 sensors-24-00639-f004:**
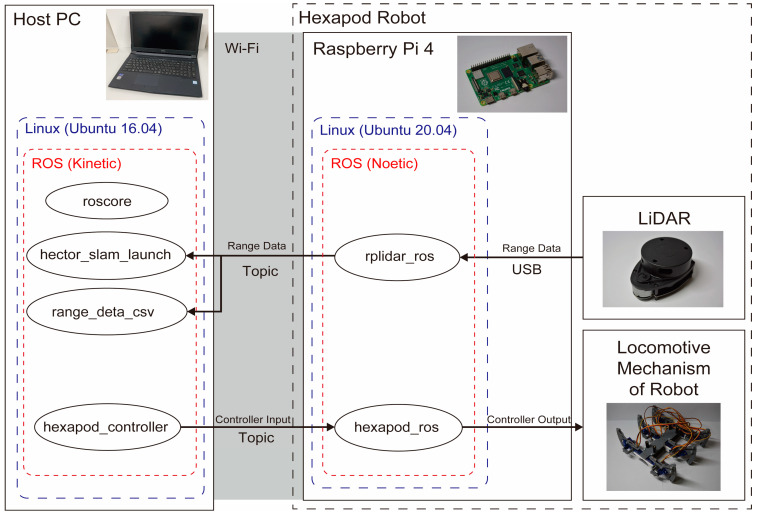
System architecture of the hexapod robot.

**Figure 5 sensors-24-00639-f005:**
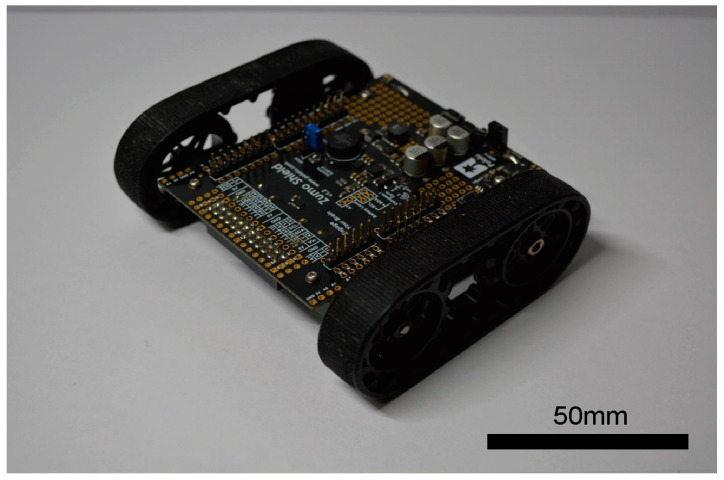
ZUMO shield for Arduino UNO3.

**Figure 6 sensors-24-00639-f006:**
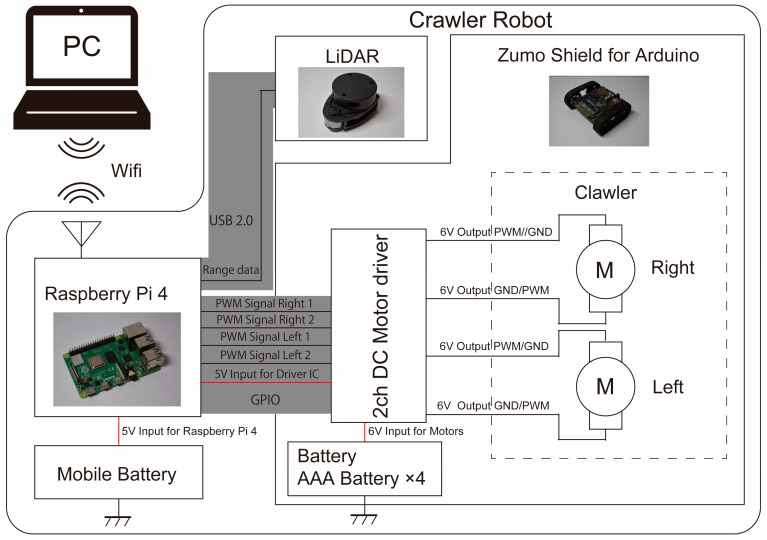
Architecture of the crawler robot.

**Figure 7 sensors-24-00639-f007:**
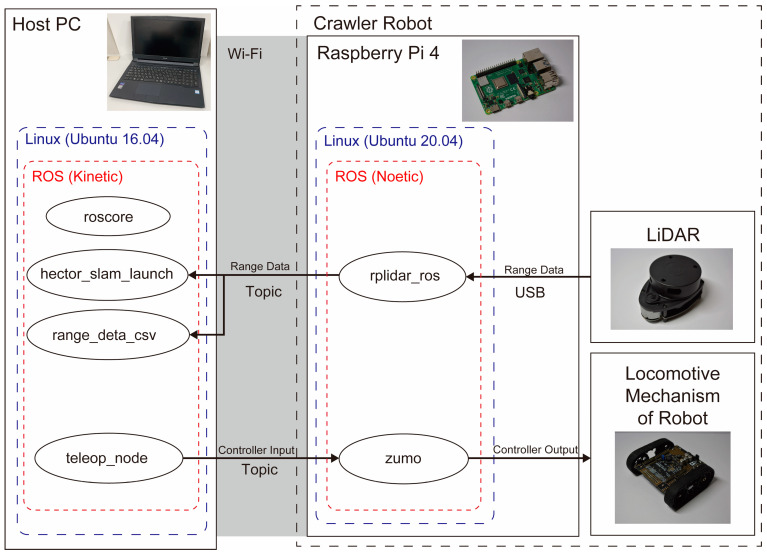
System architecture of the crawler robot.

**Figure 8 sensors-24-00639-f008:**
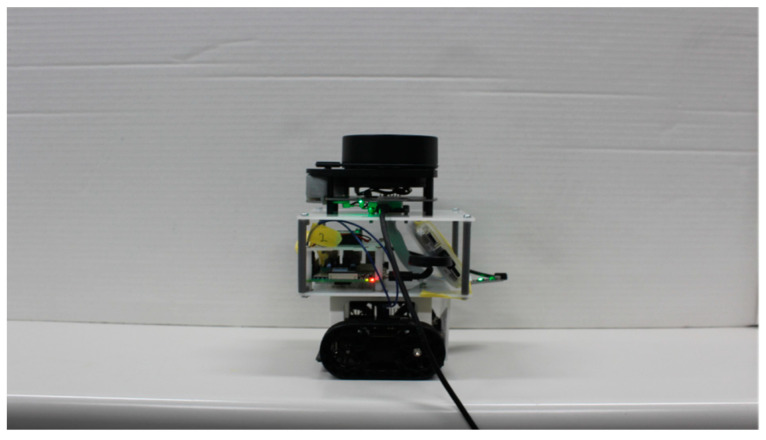
Experimental environment for pose estimation.

**Figure 9 sensors-24-00639-f009:**
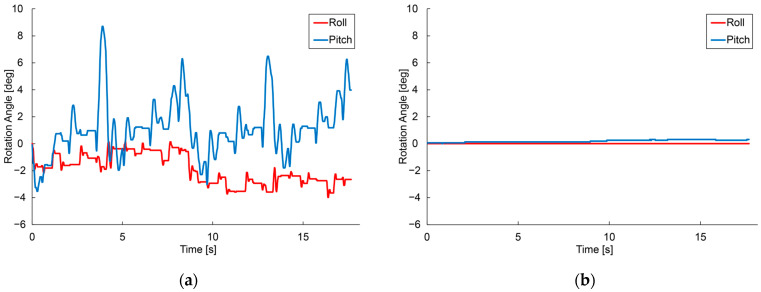
Robot Body Axis Changes. (**a**) Hexapod robot. (**b**) Crawler robot.

**Figure 10 sensors-24-00639-f010:**
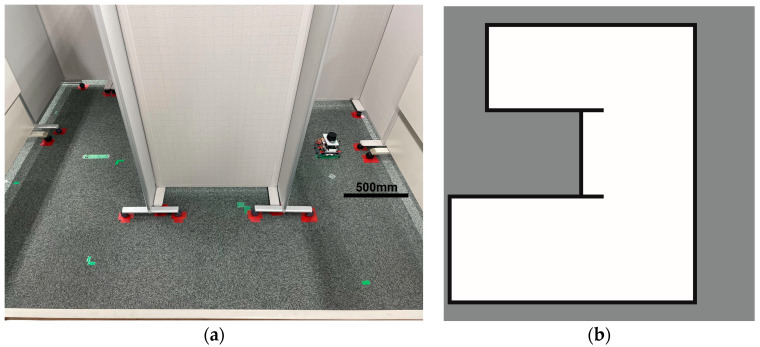
Experimental environment for simultaneous localization and mapping (SLAM). (**a**) Actual experimental environment. (**b**) Map of the experimental environment (ground truth).

**Figure 11 sensors-24-00639-f011:**
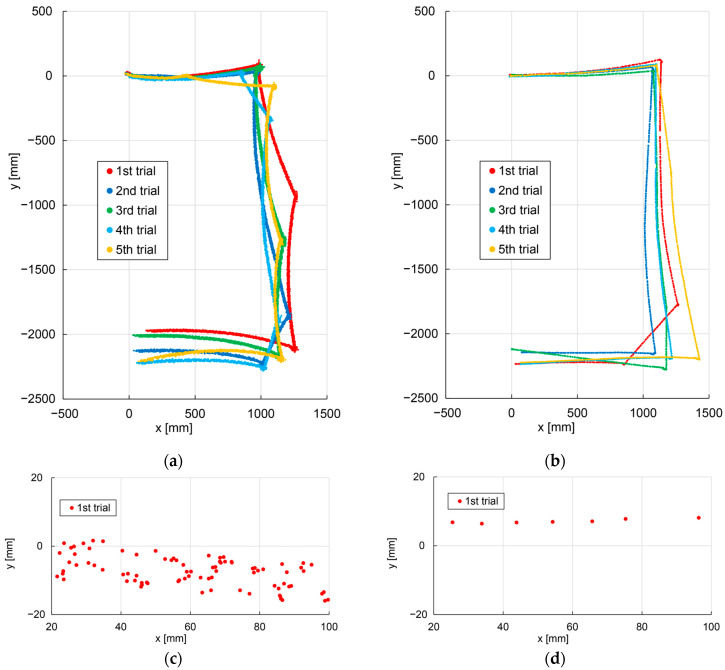
Traveled Path. (**a**) Hexapod robot. (**b**) Crawler robot. (**c**) Hexapod robot first trial (enlarged). (**d**) Crawler robot first trial (enlarged).

**Figure 12 sensors-24-00639-f012:**
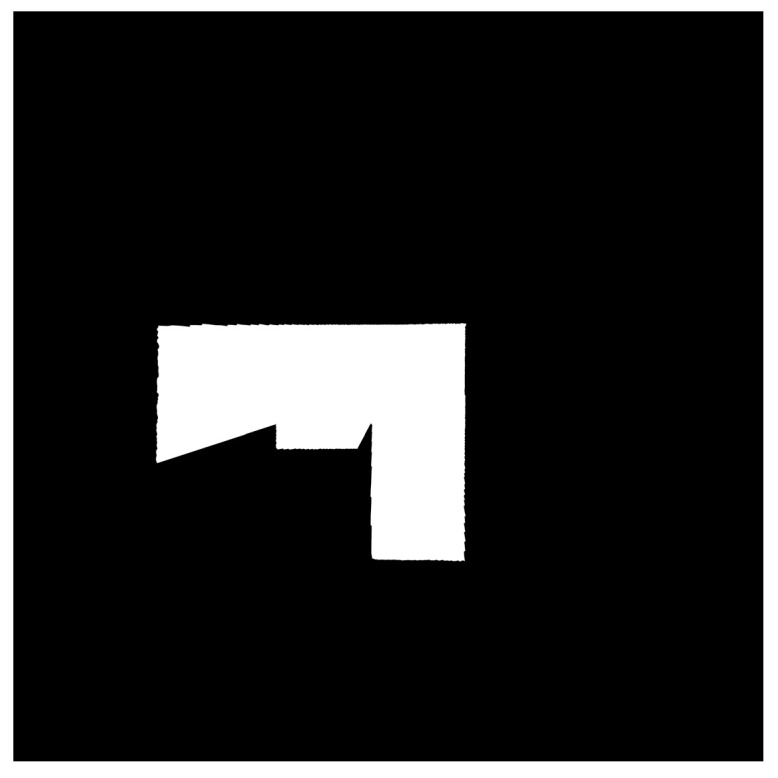
Connecting adjacent points in LiDAR point cloud data with a straight line.

**Figure 13 sensors-24-00639-f013:**

Calculation of the similarity of temporally adjacent LiDAR point cloud data.

**Figure 14 sensors-24-00639-f014:**
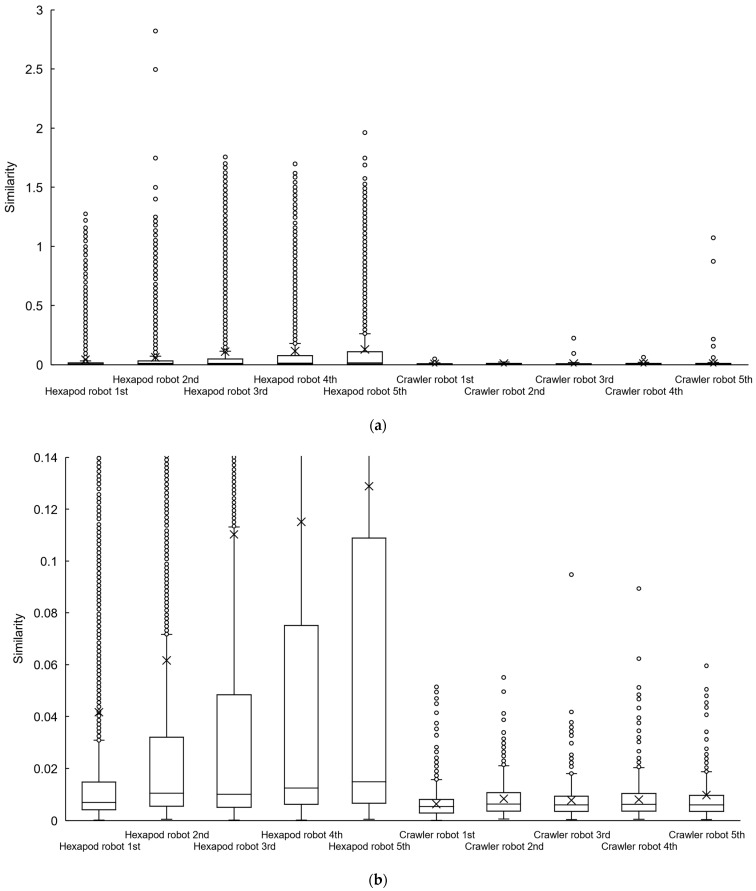
(**a**) Similarity of point cloud data adjacent in time for each trial. (**b**) Enlarged version of the same figure.

**Figure 15 sensors-24-00639-f015:**
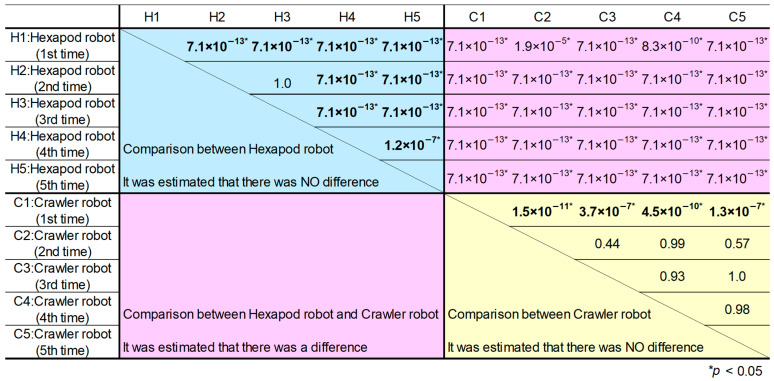
Significant difference test between 10 groups, determined using the Steel–Dwass method.

**Table 1 sensors-24-00639-t001:** Specifics of the robot.

Spec	Hexapod Robot	Crawler Robot
Size (W × H × L) (mm)	205 × 183 × 220	100 × 168 × 120
Weight (g)	1020	736
Speed (mm/s)	7	100

**Table 2 sensors-24-00639-t002:** Image of the generated map and the similarity between the generated map and the ground truth.

Number of Trials	Hexapod Robot	Crawler Robot
1st	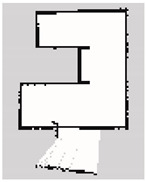	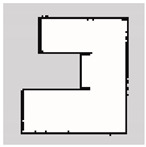
	0.0746	0.0764
2nd	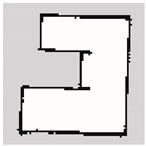	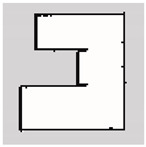
	0.0819	0.0618
3rd	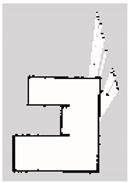	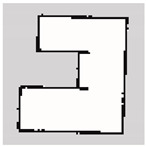
	0.0966	0.0717
4th	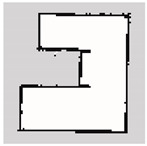	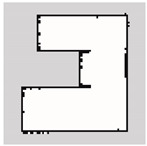
	0.0995	0.0609
5th	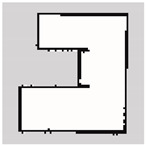	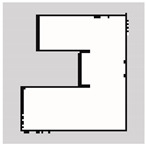
	0.0973	0.0676
Average	0.0900	0.0677

## Data Availability

The data presented in this study are available in the insert article.

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
