# Peer review of "The Influence of Micro-Hexapod Walking-Induced Pose Changes on LiDAR-SLAM Mapping Performance"

_sensors, 2024, doi:10.3390/s24020639_

Round 1

Reviewer 1 Report

Comments and Suggestions for Authors

    1. Most of the references in this paper are somewhat too old.

 2. It is best to add a paragraph at the beginning of the article to introduce each parts of the article.

    3. In Table 1, the movement speed of the robot should be supplemented.

    4. The title of 3.1 is changed to better.

    5. The article is not rigorous enough. In the summary section, there is a sentence, "possibly due to body axis changes resulting from the walking motion of the robot," and you should conduct relevant experiments to prove it, rather than simply guess it.

  6. Supplementary experiments should be performed to demonstrate the impact of changes in different body axes on positioning accuracy.

Reviewer 2 Report

Comments and Suggestions for Authors

Interesting to see that SLAM can be performed in well small platforms, which are also not operating in friendly conditions (such as using wheels on flat-horizontal terrains.

However, the paper should clearly show and focus on (and give more details) operation on terrains that are more complicated than those shown in the experimental section.

For instance, in the provided example we can see it is mostly flat-horizontal, and that there are few salient objects (base of separators/walls); however, there is no indication about if the path does pass over those salient objects (i.e. the travelled path is never presented)

In addition, a small section, showing the results in a context a bit more difficult would be realistic to be consistent with the usual difficult cases found in search and rescue  applications, for which small hexapods would be an excellent solutions.

Also, more details about the 3D attitude variations (just some plots attitude against time) would be also good indication of how challenging the robot trip may have been.

(A small IMU may help for that, or, if 3D SLAM is implemented, the 3D Attitude can be estimated as well)

So, I would recommend the authors include those extensions to the work.

Comments on the Quality of English Language

Minor refining/proofreading would be convenient:

Such as in the sentence:  “research……. has been lacking…”  

Has been lacking what?

Round 2

Reviewer 2 Report

Comments and Suggestions for Authors

The paper is now adequate for being published.

The authors have not strictly agreed with one of my recommendations, but I accept the answer provided by them, as the paper did not focus on what I had mentioned. Yet, the paper covers well interesting aspects of the problem.